# Economic evaluation of tranexamic acid for the treatment of acute gastrointestinal bleeding: a cost-effectiveness analysis using data from the HALT-IT randomised controlled trial

Nuha Bazeer,[1,2] Alec Miners,[3] Ian Roberts,[4] Haleema Shakur-Still,[4] Vipul Jairath ![ORCID],[5] Jack Williams ![ORCID] [3]

For numbered affiliations see end of article.

**Correspondence to**
Jack Williams;
jack.williams@lshtm.ac.uk

## ABSTRACT

**Objective** To perform an economic evaluation of tranexamic acid (TXA) versus no-TXA, in addition to current clinical practice, for acute gastrointestinal bleeding, using the results of the HALT-IT trial (NCT01658124), a large randomised controlled trial which included 11 937 patients.

**Design** A cost-effectiveness modelling analysis, performed over a lifetime time horizon.

**Setting** The analysis was performed from a UK health service perspective.

**Participants** The model includes adults with acute gastrointestinal bleeding.

**Outcomes measures** The model reports costs in Great British pounds in 2021 and outcomes as life years (LYs) and quality-adjusted life years (QALYs). Cost-effectiveness was evaluated using incremental cost-effectiveness ratios (ICERs), reported as the cost per QALY gained.

**Methods** A Markov model was developed to calculate the overall costs and health outcomes of TXA administration versus no-TXA. The model used data of the treatment effectiveness from the HALT-IT trial, which showed that TXA administration for acute gastrointestinal bleeding did not reduce all-cause mortality (risk ratio 1.03, 95% CI 0.92 to 1.16) compared with no-TXA. Data on health-related quality of life, costs and long-term mortality risks were derived from the literature. Costs and effects are discounted at 3.5% per annum.

**Results** TXA was associated with marginally fewer LYs and QALYs, and lower costs, than treatment without TXA. The ICER associated with no-TXA was £1576 per LY gained and £2209 per QALY gained. No-TXA was 64% likely to be cost-effective at a £20 000 willingness-to-pay threshold, while TXA was 36% likely to be cost-effective.

**Conclusion** Though inexpensive, TXA administration for patients with acute gastrointestinal bleeding is unlikely to be cost-effective.

## STRENGTHS AND LIMITATIONS OF THIS STUDY

⇒ This is the first economic evaluation of tranexamic acid for the treatment of acute gastrointestinal bleeding.

⇒ Data for the treatment effect comes from the HALT-IT trial, which included 11 937 patients receiving either tranexamic acid or placebo.

⇒ The cost-effectiveness model extrapolates the impact of tranexamic acid treatment over a lifetime time horizon, as recommended by the National Institute for Health and Care Excellence, for UK based economic evaluations.

⇒ The HALT-IT trial data only reports outcomes up to 28 days following the onset of acute gastrointestinal bleeding, and therefore the model only accounts for treatment effects observed within this 28-day trial period.

## INTRODUCTION

Acute gastrointestinal bleeding (AGIB) refers to bleeding arising from the upper or lower gastrointestinal tract. Acute upper gastrointestinal bleeding (AUGIB) occurs more commonly than acute lower gastrointestinal bleeding (ALGIB) and has a higher case fatality rate of up to 10%.[1] AUGIB is a common medical emergency and results in approximately 85 000 cases each year within the UK, with around 4000 AUGIB-related deaths annually.[2] Thus, AUGIB presents a significant health concern with an estimated incidence of 133 cases per 100 000 population, accounting for 8% of all acute hospital admissions.[2] The leading causes of AUGIB have been recorded as peptic ulcers and oesophagogastric varices.[1 3] While peptic ulcers are often associated with long-term use of non-steroid anti-inflammatory drugs,

oesophagogastric varices are typically complications of liver cirrhosis.[4] Comparisons of national AUGIB audits in the UK carried out in 1994 and 2007 have shown improved outcomes among patients, likely through advancements in endoscopic and some pharmacological therapies.[1] However, over the past two decades, there have been no significant changes to mortality associated with AUGIB, and mortality among older patients with other comorbidities remains high at 10%.[2 5] Within the UK, annual in-hospital costs for all AUGIB cases have been estimated as £155.5 million, of which £12.6 million accounts for blood product transfusions (BPT) alone.[6] As the second most common medical reason for transfusion, gastrointestinal (GI) bleeding accounts for 14% of all BPT within the UK, with inappropriate use recorded in 20% of cases.[7 8]

Tranexamic acid (TXA) is an antifibrinolytic agent which reduces fibrinolysis and inhibits the breakdown of blood clots.[9] It is used to treat women with heavy menstrual bleeding (menorrhagia) and people with haemophilia during dental extractions.[10–13] The efficacy of TXA has also been demonstrated in reducing blood loss during surgical operation trials.[9] It is widely available, relatively inexpensive and can be administered through oral or intravenous formulations, without specialist monitoring.[14] Large, multicentre, randomised controlled trials (RCTs) such as the WOMAN and CRASH-2 and CRASH-3 trials have shown TXA to significantly reduce bleeding mortality among patients at risk of postpartum haemorrhage and bleeding trauma patients, respectively.[15–17] However, the timing of TXA administration is of particular importance, with no evidence of a benefit on mortality when administered more than 3 hours after injury in both WOMAN and CRASH-2 and CRASH-3 trials.[16 18] Unlike postpartum haemorrhage or brain injury however, the onset of a gastrointestinal bleed may be less discernible, with presentation often delayed due to symptoms only becoming visible when most severe. This was observed within the HALT-IT trial, with over 80% of patients presenting more than 3 hours after the onset of bleeding.[19]

Previously conducted smaller RCTs in the UK have found no evidence of a difference in mortality, rebleeding and the requirement of surgery.[20 21] However, a 2014 Cochrane systematic review and meta-analysis of RCTs comparing TXA with placebo for AUGIB estimated large reductions in all-cause mortality, with a pooled risk ratio (RR) of 0.6 (95% CI 0.42 to 0.87) across eight studies with 1701 participants.[22] However, the mortality benefit of TXA was uncertain when excluding trials with a high risk of bias. For this reason, the authors concluded that TXA cannot be recommended for routine clinical practice. An updated systematic review and meta-analysis found similar findings, with a RR of 0.59 (95% CI 0.43 to 0.82), and also concluded that high-quality RCTs were needed.[23]

TXA is not currently recommended within UK guidelines for AUGIB or ALGIB.[24] National Institute for Health and Care Excellence (NICE) guidelines for AUGIB refer to limited evidence of an effect of TXA on mortality, but do not include recommendations for TXA in the absence of UK marketing authorisation for this indication. Similarly, guidelines from the British Society of Gastroenterology state that the mortality benefit of TXA is uncertain when excluding trials with a low risk of bias, suggesting recommendations are made following the HALT-IT trial.[25]

The HALT-IT trial (NCT01658124) was a large multinational RCT that compared TXA versus placebo, in addition to current practice, in 11 937 people with AGIB.[19] The trial protocol has also been published.[26] The trial did not find evidence that TXA significantly reduces any of the key outcome measures, such as death due to bleeding within 5 days of randomisation (RR 0.99, 95% CI 0.82 to 1.18) or all-cause mortality (RR 1.03, 95% CI 0.92 to 1.16). This means by default, even if relatively inexpensive, TXA is unlikely to represent value for money in this indication. However, there is still a need to perform an economic evaluation based on the HALT-IT results, in order to establish the level of uncertainty about this conclusion and to assess the extent to which only including the results from the systematic review would have impacted the cost-effectiveness estimates. The full results of the HALT-IT trial, including the economic results, are available in a separate National Institute of Health Research report.[27]

## METHODS

### Analysis

The base case economic analysis assessed the cost-effectiveness of treating patients identified as sustaining an AGIB with TXA versus no-TXA, in addition to current practice, using estimates of treatment effect from the HALT-IT trial. In line with NICE guidelines for health technology appraisals, a cost-effectiveness model was developed in Microsoft Excel and was analysed over a lifetime time horizon, from a UK National Health Service (NHS) health service perspective.[28] All costs are reported in 2021 Great British pounds, with costs inflated using an NHS cost inflation index.[29] Health outcomes were assessed as life years (LYs), and quality-adjusted life years (QALYs). All future costs and health outcomes were discounted at 3.5% per annum, as per NICE guidance.[28] We report the results as incremental cost-effectiveness ratios (ICERs) by dividing the incremental costs by the incremental outcomes. We also report results using incremental net monetary benefits (INMBs), by converting incremental outcomes in monetary values based on the willingness-to-pay threshold, and then subtracting the incremental costs. This means that positive INMB represents a cost-effective scenario, while a negative INMB does not. A willingness-to-pay threshold of £20 000 per QALY has been used to determine cost-effectiveness, in line with NICEs Technology Appraisals recommendations.[28]

The HALT-IT trial included a number of subgroup analyses, but as no treatment effects were observed among subgroups, they were not considered in the model. While

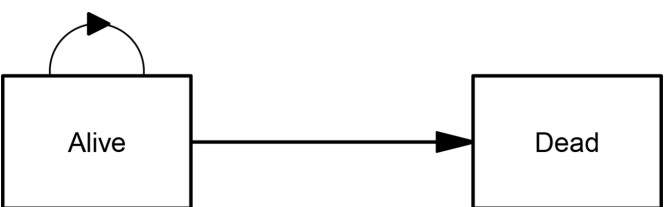

**Figure 1** Markov model structure showing both health states within the model.

the trial consisted mostly of patients with AUGIB (89%), the model population includes both AUGIB and ALGIB, as per the overall trial population.

## Model structure

An existing Markov model of TXA following traumatic brain injury was adapted to estimate the cost-effectiveness of TXA for patients with AGIB.[30] The model consists of two health states *alive* and *dead*; as shown in figure 1. People with AGIB enter the model in the *alive* health state. They progress to the *dead* health state over the initial 28-day period (the HALT-IT trial period) using daily data taken directly from the trial on all-cause deaths from the placebo arm of HALT-IT; this includes data for all UK and non-UK participants. For the remaining 337 days of the first year, the daily probability of death in the no-TXA arm was estimated using UK general population age-related mortality statistics,[31] adjusted using a standardised mortality ratio (SMR) to reflect the long-term risk of excess death associated with surviving an AGIB.[32] In the base case, this SMR is applied for the entire time horizon. After this initial year period, the model uses an annual cycle length, and applies a probability of death based on UK all-cause mortality data,

combined with the SMR. In line with the HALT-IT trial, the mean age of people entering the model was assumed to be 58.1 years (table 1).

The SMR associated with the risk of death following an AGIB compared with that of the general population was derived from a study by Crooks *et al*.[32] It linked longitudinal data from the UKs Hospital Episodes Statistics data set, General Practice Research Database and Office of National Statistics death register for people with non-variceal bleeds between 1997 and 2010. Controls were matched by age, sex, practice and year. The study reported the risk of death was highest in the first year following an AGIB (SMR 5.21, 95% CI 4.25 to 6.38), and remained elevated from year two onwards (SMR 1.74, 95% CI 1.42 to 2.13).

## Treatment effect

The HALT-IT trial results for the primary outcome measure, death due do bleeding within 5 days from randomisation, was similar in both treatment arms (RR 0.99, 95% CI 0.82 to 1.18), although the mean estimate slightly favoured TXA.[19] However, as the risks of venous thromboembolic events (RR 1.85, 95% CI 1.15 to 2.98), death from malignancy (RR 1.63, 95% CI 1.10 to 2.42), and seizure (RR 1.73, 95% CI 1.03 to 2.93) were higher in the TXA-group than placebo-group, the base case RR of all-cause mortality was used to estimate the overall effect of TXA treatment on survival (RR 1.03, 95% CI 0.92 to 1.16), and is shown in table 1. This RR is based on all trial participants and is not UK specific since the subgroup analysis did not reveal any differences in treatment effect by country income level.

| Table 1 | Base case model input parameters | | | |
|---|---|---|---|---|
| **Parameter** | **Value** | **95% CIs** | **Distribution** | **Source** |
| Age at model start | *58.1* | – | – | HALT-IT |
| SMR year 1 | *5.21* | 4.25 to 6.38 | Log normal | Crooks *et al*[32] |
| SMR year 2 | *1.74* | 1.42 to 2.13 | Log normal | Crooks *et al*[32] |
| RR of all-cause mortality | *1.03* | 0.92 to 1.16 | Log normal | HALT-IT |
| 28-day probability of death* | *0.092* | 0.084 to 0.099 | Beta | HALT-IT |
| Utility AGIB survivor | *0.735* | 0.70 to 0.77 | Beta | Campbell *et al*[6] |
| Utility decrements by age (years) | | | | Kind *et al*[35] |
| 55–64 | *0* | – | – | |
| 65–74 | *0.02* | – | – | |
| ≥75 | *0.07* | – | – | |
| Utility UK general population | *By age* | – | – | Kind *et al*[35] |
| All-cause mortality | *By age* | – | – | ONS[31] |
| Discount rate costs | 3.5% | – | – | NICE[28] |
| Discount rate QALYs | 3.5% | – | – | NICE[28] |

*The 28-day risk of death in the placebo arm was sampled from a beta distribution during the probabilistic sensitivity analysis, based on the proportion of deaths during the trial period (548/5981). The deaths occurring on each day, as a proportion of the overall 28-day risk, remained the same during sampling.

AGIB, acute gastrointestinal bleeding; QALYs, quality-adjusted life years; RR, risk ratio; SMR, standardised mortality ratio.

## Utilities

Utility data were not collected as part of HALT-IT, therefore values were sourced from the existing literature. Only one directly relevant study could be identified—the relatively recent TRIGGER RCT.[33] The EQ-5D-3L (EuroQol 5-dimensions 3-levels) questionnaire was administered at discharge or day-28, with an estimated utility of 0.735 (95% CI 0.70 to 0.77) for those surviving an AGIB (table 1).[6] In line with previous economic evaluations of TXA, this single utility value of 0.735 was applied to all people who were alive at day 28, both during the initial 28-day period and at all times thereafter until death, in both treatment arms.[30] Note that functional status was measured in HALT-IT using the Katz Index of Independence in Activities of Daily Living,[34] with the scores similar in both arms (score of 5.5 for both).

Since the TRIGGER RCT population has a slightly lower level of comorbidities and complications among participants compared with the HALT-IT population, we considered the impact of a much lower utility value (0.5) for all surviving patients following AGIB.

An age adjustment was also applied to the utility value each cycle using the decrements reported by Kind et al, so that overall values declined with age.[35]

## Costs

### Treatment costs

The resource use associated with providing TXA treatment were derived from the HALT-IT trial and by making a number of assumptions. The total TXA dose (4 g) included a loading-dose and maintenance-dose, for which the total cost was derived from the British National Formulary (£12 per patient).[36] The cost of equipment for treatment was assumed to include a needle (£0.05), syringe (£0.07), sodium chloride infusion bag (£0.59) and isotonic intravenous solution (£2.96).[36] To account for the administration time of TXA, hourly staff costs were taken from the Personal Social Services Research Unit for 2021.[29] The model assumed that a Band 5 nurse would take 21 min to administer TXA (£14.35), following the treatment administration and expert guidance used within the economic evaluations of the CRASH[30 37] and WOMAN trials.[38] This led to a total TXA treatment and administration cost of £30.01 per person (table 2).

### Inpatient stay costs

Information from HALT-IT was used to derive the costs of inpatient stays during the initial 28-day trial period. The total mean lengths of stay recorded by people receiving TXA and placebo were 5.83 and 5.80 days, respectively (table 2). In both treatment arms, 0.4 of these days were spent in intensive care units. These lengths of stay were multiplied by NHS Reference Costs for non-elective stays to derive mean hospital costs for TXA and no-TXA of £2471 and £2462 per person, respectively.[39]

### Procedures and transfusions

While in hospital during the 28-day trial period, participants underwent a number of procedures, including endoscopy, surgery and radiological interventions (table 2). The cost of these procedures were calculated for each trial arm using the probability of occurrence from the HALT-IT trial multiplied by the NHS Reference Costs.[39] The mean per person procedural costs for the TXA and no-TXA treatment options were £2389 and £2437, respectively.

People also received transfusions of blood/red blood cells, fresh frozen plasma and platelets while in hospital (table 2). Information from HALT-IT on the frequency and the mean number of units for each transfusion type were combined with unit costs from the NHS Blood and Transfusions price list to generate overall costs.[40] The mean per person transfusion costs for the TXA and no-TXA treatment options were £253 and £264, respectively.

While the model did not explicitly account for adverse events during the trial period, the costs of these adverse events are included in the hospital length of stay, procedures and transfusions that occur during the hospital stay.

### Post discharge costs

The HALT-IT and TRIGGER trials observed participants for a maximum of 28-days, therefore longer-term care costs were sourced from the literature. A specific study related to AGIB could not be identified. However, a recent UK study by Ramagopalan et al quantified the costs of GI bleeding among people with non-valvular atrial fibrillation, who were and were not experiencing a bleed, and used a difference-in-difference approach to calculate the costs attributable to a GI bleed (cases, n=7753 and controls, n=7753).[41]

The study used NHS reference costs and calculated the annual costs of caring for people with a bleed to be £4350, £1980 and £1938 in the first, second and third years postbleed, respectively (table 2). In the absence of other information, the reported year 3 costs were inputted into the model for year 4 onwards, although they were assumed to be one-third of the original amount. The SEs that the costing study reports could not be easily incorporated into our analysis (the statistical approach used allowed costs to be negative), therefore they were assumed to be 50% of the mean value in all instances, which is thought to reflect a relatively large degree of uncertainty around the mean estimates. The importance of these assumptions was assessed in the sensitivity analysis.

### Sensitivity analysis

Various one-way deterministic sensitivity analyses (DSA) were undertaken to assess the robustness of the results to alternative assumptions. In line with NICE guidelines, one-way DSA was conducted on the discount rate for both costs and health outcomes, using values of 0% and 6%, respectively. It was also conducted on a range of scenarios considering the extrapolation of the monitoring costs,

**Table 2**  Base case cost parameters

| Parameter | Value | SE or proportion | Distribution | Source |
|---|---|---|---|---|
| TXA administration | £30.01 | – | – | BNF[36], PSSRU[29], assumption |
| Inpatient stays to day-28 | | | | |
| Total length of stay (days) | | | | |
| TXA | 5.83 | 1.46 | Gamma | HALT-IT |
| No-TXA | 5.80 | 1.47 | Gamma | HALT-IT |
| Days in ICU | | | | |
| TXA | 0.4 | 0.46 | Gamma | HALT-IT |
| No-TXA | 0.4 | 0.51 | Gamma | HALT-IT |
| Unit cost per day non-ICU | £338 | – | – | Reference costs[39] |
| Unit cost per day on ICU | £1594 | – | – | Reference costs[39] |
| Procedures to day 28 | | | | |
| Prob endoscopy - diagnostic | | | | |
| TXA | 0.8 | 4781/5953 | Beta | HALT-IT |
| No-TXA | 0.79 | 4729/5978 | Beta | HALT-IT |
| Unit cost | £665 | – | – | Reference costs[39] |
| Prob endoscopy - therapeutic | | | | |
| TXA | 0.43 | 2542/5953 | Beta | HALT-IT |
| No-TXA | 0.44 | 2658/5978 | Beta | HALT-IT |
| Unit cost | £777 | – | – | Reference costs[39] |
| Prob surgical intervention | | | | |
| TXA | 0.02 | 146/5953 | Beta | HALT-IT |
| No-TXA | 0.03 | 158/5978 | Beta | HALT-IT |
| Unit cost | £1377 | – | – | Reference costs[39] |
| Prob radiological - diagnostic | | | | |
| TXA | 0.29 | 1704/5953 | Beta | HALT-IT |
| No-TXA | 0.29 | 1744/5978 | Beta | HALT-IT |
| Unit cost | £4986 | – | – | Reference costs[39] |
| Prob radiological - therapeutic | | | | |
| TXA | 0.01 | 74/5953 | Beta | HALT-IT |
| No-TXA | 0.01 | 89/5978 | Beta | HALT-IT |
| Unit cost | £4986 | – | – | Reference costs[39] |
| Transfusions | | | | |
| Prob blood or red cells | | | | |
| TXA | 0.67 | 3984/5951 | Beta | HALT-IT |
| No-TXA | 0.67 | 4018/5978 | Beta | HALT-IT |
| Mean units of blood or red cells | | | | |
| TXA | 2.80 | 0.61 | Gamma | HALT-IT |
| No-TXA | 2.90 | 0.69 | Gamma | HALT-IT |
| Unit cost | £132 | – | – | NHS Blood and Transport[40] |
| Prob fresh frozen plasma | | | | |
| TXA | 0.15 | 910/5951 | Beta | HALT-IT |
| No-TXA | 0.16 | 993/5978 | Beta | HALT-IT |
| Mean units fresh frozen plasma | | | | |
| TXA | 0.90 | 0.61 | Gamma | HALT-IT |
| No-TXA | 1.0 | 0.66 | Gamma | HALT-IT |

**Table 2** Continued

| Parameter | Value | SE or proportion | Distribution | Source |
|---|---|---|---|---|
| Unit cost | £31 | – | – | NHS Blood and Transport[40] |
| Prob platelets | | | | |
| TXA | 0.04 | 219/5951 | Beta | HALT-IT |
| No-TXA | 0.04 | 255/5978 | Beta | HALT-IT |
| Mean units of platelets | | | | |
| TXA | 0.20 | 0.23 | Gamma | HALT-IT |
| No-TXA | 0.20 | 0.26 | Gamma | HALT-IT |
| Unit cost | £192 | – | – | NHS Blood and Transport[40] |
| Post discharge costs | | | | |
| Year 1 | £4350 | £2175* | Gamma | Ramagopalan et al[41] |
| Year 2 | £1980 | £990* | Gamma | Ramagopalan et al[41] |
| Year 3 | £1938 | £969* | Gamma | Ramagopalan et al[41] |

*SEs were assumed to be 50% of the mean value in the base case analysis.
ICU, intensive care unit; TXA, tranexamic acid.

health-related quality of life for AGIB survivors and TXA treatment effect.

To simultaneously investigate the effect of parameter uncertainty, probabilistic sensitivity analysis (PSA) was also undertaken using 10 000 Monte Carlo simulations; the parameter distributions are specified in table 1 and table 2. The probabilistic results are presented on a cost-effectiveness plane, and as a cost-effectiveness acceptability curve (CEAC). Parameters that were excluded from the PSA are not assigned a distribution type in the tables.

A scenario analysis was performed to consider the economic results that would have been generated prior to the availability of the results of TXA from the HALT-IT trial. This analysis was performed primarily to demonstrate the differences in decision-making when using low or moderate quality evidence to populate the economic model, particularly from trials with a high risk of bias. In this analysis the 28-day mortality risk was 0.084 (71/850) in the control group, while the TXA all-cause mortality RR was 0.60 (95% CI 0.42 to 0.87), as per the Cochrane analysis.[22]

### Patient and public involvement

Patient and public input was used to inform the HALT-IT trial procedures, including study information sheets and consent forms.[27] Patients stated that the research should account for quality of life issues in additional to the treatment efficacy and safety, which this economic analysis does. There was no patient or public involvement specifically in the design of the economic model.

### RESULTS
### Base case analysis

The base case results are shown in table 3. In both the deterministic and probabilistic analyses TXA was associated with marginally fewer LYs and QALYs, and marginally lower costs, than treatment without TXA. The deterministic QALYs and costs associated with TXA and no-TXA were 8.58 and 8.61, and £18 155 and £18 220, respectively. This meant that providing TXA was £64.30 less costly, and resulted in 0.041 fewer QALYs, per person treated.

Since TXA was less costly and less effective in the base case analysis compared with placebo, the ICERs can be reported as £1576 saved for each LY lost, and £2209 saved for each QALY lost. Since we value a QALY at £20 000 using the NICE willingness-to-pay threshold, this does not represent a cost-effective option, because the value of the health outcomes foregone are higher than the cost savings which occur. Alternatively the results can be expressed as no-TXA versus TXA, with no-TXA costing £1576 per LY gained and £2209 per QALY gained, meaning no-TXA is

**Table 3** Base case deterministic results, including a breakdown of cost components

| | No-TXA | TXA |
|---|---|---|
| Total costs | £18 220 | £18 155 |
| TXA costs | £0 | £30 |
| Procedures and transfusions | £5162 | £5112 |
| Monitoring costs | £13 057 | £13 013 |
| Life years | 12.08 | 12.04 |
| QALYs | 8.61 | 8.58 |
| Incremental costs | | –£64.30 |
| Incremental QALYs | | –0.041 |
| ICER (LY) | | £1576 |
| ICER (QALY) | | £2209 |

ICER, incremental cost-effectiveness ratio; LY, life year; QALYs, quality-adjusted life years; TXA, tranexamic acid.

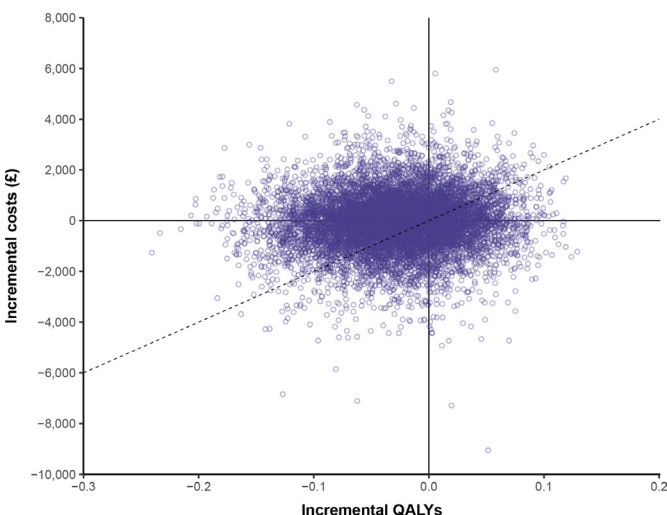

**Figure 2** Base case cost-effectiveness plane. The dotted line represents the £20 000 willingness-to-pay threshold; points falling on or below this line are considered cost-effective. The NE quadrant contains 15% of simulations; SW quadrant 38%; SE quadrant 14%; NW quadrant 33%. QALYs, quality-adjusted life years.

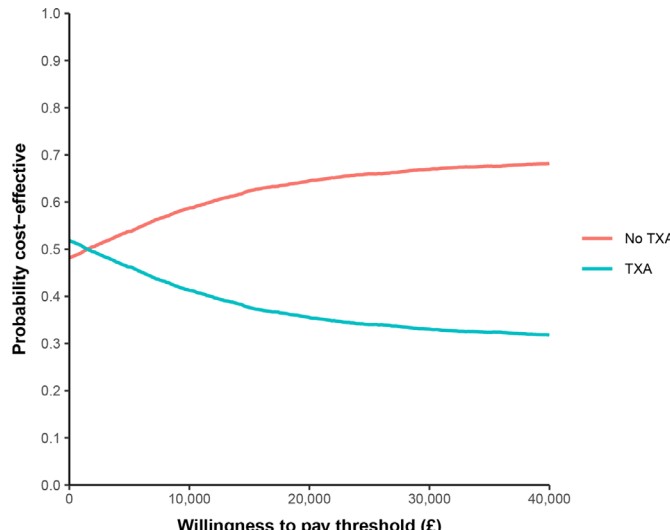

**Figure 3** Base case cost-effectiveness acceptability curve. TXA, tranexamic acid.

preferred at the NICE willingness-to-pay threshold. Either way, providing TXA is not cost-effective. The deterministic and mean probabilistic results are very similar.

The lower mean health outcomes for TXA are driven by the fact the mean RR for all-cause mortality (ie, the relative treatment effect) slightly favours the no-TXA treatment arm (RR 1.03). The higher incremental costs associated with no-TXA administration are largely driven by the fact people who did not receive TXA live for marginally longer than those who received TXA, thus they incur slightly higher post discharge costs, and marginally higher costs for procedures and transfusions, exceeding the additional low cost of TXA administration.

### Probabilistic sensitivity analysis

The cost-effectiveness plane shows that the PSA simulations are clustered around the origin and distributed across all four cost-effectiveness quadrants (figure 2). This is because the point estimate for the RR of all-cause mortality is only slightly above 1, and is included within its CI (RR 1.03, 95% CI 0.92 to 1.16). Therefore, there is little to choose between the two treatment options in terms of outcomes. The same is also true of the relatively low cost of TXA administration (£30.01 per person), meaning that the majority of the costs incurred are the hospital costs (hospital stay, procedures and transfusions) and post-discharge costs.

The CEAC (figure 3) shows that at any willingness-to-pay for an additional QALY above £2000, no-TXA is likely to be the more cost-effective option. At the NICE specified willingness-to-pay threshold levels of £20 000 and £30 000 per additional QALY, TXA has a 36% and 33% probability of being cost-effective, respectively. Note that the base case CEAC does not indicate that there is a 0% probability TXA is cost-effective, irrespective of

the willingness-to-pay value. TXA treatment produced more QALYs at lower cost (ie, it was the 'dominant' treatment option) in 15% of simulations. Rerunning the PSA assuming the post discharge costs had a SE of 20%, instead of 50%, of their mean value, had a negligible impact on the probability of TXA being cost-effective.

### Deterministic sensitivity analysis

Various deterministic one-way sensitivity analyses are shown table 4. In all but one instance the results show that TXA is not cost-effective, and the ICER was very robust to alternative model parameter and structural assumptions. The exception is when the RR of all-cause mortality is reduced to 0.92 (the lower bound of the 95% CI). In this instance, TXA produces marginally better health outcomes at slightly higher costs than no-TXA, producing an ICER of £1227 per QALY gained if TXA is used instead of no-TXA. Note that even when the cost of TXA and its administration was set to £0 instead of the base case value of £30.01, it is not cost-effective.

### Meta-analysis scenario analysis

When parameterising the model with the mortality risk and all-cause mortality treatment effect reported from a meta-analysis containing multiple studies at risk of bias, prior to the availability of the HALT-IT trial, TXA becomes highly cost-effective, with an ICER of £1040 per LY gained and £1459 per QALY gained. In the probabilistic analysis, TXA was 99.5% likely to be cost-effective at the £20 000 per QALY threshold.

### DISCUSSION

The results from the base case analysis suggest that the costs and outcomes of treating people with AGIB with and without TXA, as enrolled and treated in the HALT-IT trial, are very similar. Despite being a relatively inexpensive intervention, TXA is unlikely to be a cost-effective

**Table 4**  Deterministic one-way sensitivity analysis results

| | Inc. costs (£)* | Inc. LY* | Inc. QALY* | ICER LY (£) | ICER QALY (£) | INMB† (QALY) (£) |
|---|---|---|---|---|---|---|
| Base case | −64 | −0.041 | −0.029 | 1576 | 2209 | −518 |
| Monitoring costs (full monitoring costs for year 4 onwards) | −106 | −0.041 | −0.029 | 2605 | 3652 | −476 |
| Monitoring costs not included beyond 3 years | −43 | −0.041 | −0.029 | 1061 | 1488 | −539 |
| 0% discount rate | −78 | −0.060 | −0.043 | 1291 | 1828 | −775 |
| 6% discount rate | −58 | −0.032 | −0.023 | 1799 | 2508 | −406 |
| Utility post discharge (0.5) | −64 | −0.041 | −0.020 | 1576 | 3297 | −326 |
| Monitoring costs - inpatient costs excluded in year 1 | −55 | −0.041 | −0.029 | 1341 | 1880 | −527 |
| TXA all-cause mortality risk ratio (0.92) | 85 | 0.097 | 0.069 | 875‡ | 1227‡ | 1304 |
| TXA all-cause mortality risk ratio (1.16) | −231 | −0.195 | −0.139 | 1185 | 1662 | −2548 |
| Total TXA administration at £0 | −94 | −0.041 | −0.029 | 2311 | 3240 | −488 |

*Negative costs, LYs and QALYs indicate that TXA is less costly and less effective than no-TXA.
†INMB calculated using a £20 000 willingness-to-pay for an additional QALY threshold.
‡In this scenario, the ICERs flip meaning they favour TXA treatment (hence the positive INMB).
ICER, incremental cost-effectiveness ratio; INMB, incremental net monetary benefit; LY, life year; QALYs, quality-adjusted life years; TXA, tranexamic acid.

use of NHS resources,[19] and no-TXA is likely to be the cost-effective option. These results are in line with our prior expectations given the additional, although modest, cost of providing TXA, which did not reduce mortality. However, the base case probabilistic analysis showed that there is a 36% chance TXA is cost-effective at £20 000 per QALY thresholds, since there was uncertainty around the all-cause mortality treatment effect, with the RR CI ranging from 0.92 to 1.16.

As with all decision models, our analysis includes a number of parameter and structural assumptions. While the analysis calculated the hospital costs associated with the hospital stay and procedures performed, the model did not explicitly include the long-term impacts of adverse events. Instead, the data used to extrapolate the costs and effects following the 28-day trial period were assumed to include patients with AGIB with similar comorbidities and complications following their GI bleed.

However, extensive sensitivity analysis showed that the parameter values for the costs and utilities following AGIB were generally unimportant in terms of changing the conclusion that TXA is unlikely to be cost-effective in this indication. The only sensitivity analysis in which TXA was cost-effective was when it was assumed to reduce mortality. Given that TXA resulted in marginally lower costs than no-TXA (due to marginally lower hospital costs during the trial period), it would be cost-effective if it resulted in any all-cause mortality benefit. Given this, if future research identified a modest treatment effect for

a subgroup of patients with AGIB, then TXA would likely be cost-effective in this group.

To the best of our knowledge, no other cost-effectiveness analyses of TXA for people with AGIB have been reported. However, economic evaluations based on the series of CRASH RCTs and WOMAN RCT have concluded TXA is highly cost-effective following trauma injury and postpartum haemorrhage.[15 16 30 38] The dose of TXA used in the HALT-IT trial is higher than that used in the previous studies, but this increased cost is negligible. The reason for these differences in the cost-effectiveness results is driven by the mortality benefit associated with TXA in other indications, which was not observed the HALT-IT trial population, or in any subgroup analyses. Of considerable importance and highlighted within the WOMAN and CRASH-3 studies was the time to treatment, with earlier treatment (within 3 hours of onset of bleeding) associated with improved outcomes.[42] In patients with AGIB, it is often difficult to identify the onset of bleeding, and presentation is often delayed. In the HALT-IT trial, 80% of participants received treatment more than 3 hours after the suspected onset of bleeding. Furthermore, patients with AGIB are older, with more comorbidities.[19]

One of the main strengths of this analysis is that it is the first economic evaluation of TXA for people with AGIB based on the results of a large RCT. The results from HALT-IT are in contrast to previous meta-analyses of small trials, in which TXA was estimated to result in

a large reduction in all-cause mortality in people with AUGIB, when including trials at risk of bias.[22 23] A scenario analysis using these meta-analysis results found that, had an economic evaluation using the results of this meta-analysis been performed prior to the HALT-IT trial, the results would have suggested TXA was almost certainly cost-effective, with little uncertainty, and indicated little value in performing additional research. With hindsight, we believe the authors of these analyses were correct to conclude that their findings were insufficient to recommend the routine use of TXA in clinical practice, since this would have potentially led to poorer outcomes at a health systems level and potentially impacted on HALT-IT recruitment. For future health economic analyses, we suggest that researchers should be cautious in their use of meta-analyses that include trials considered susceptible to error or bias, which could subsequently result in misleading policy recommendations.

In carrying out an economic evaluation with robust sensitivity analysis, our findings emphasise the importance of large RCTs designed to ascertain the effectiveness of an intervention, and the importance of an economic analysis using high quality RCT data, to assess the cost-effectiveness of low-cost interventions.

## CONCLUSION

Building on the evidence provided from the largest RCT in patients with AGIB, the results of this analysis have shown TXA administration for patients with AGIB is, on balance, unlikely to be cost-effective at UK willingness-to-pay thresholds, and should not be recommended based on the current economic evidence. However, there remains uncertainty around this conclusion, with TXA being cost-effective in approximately one-third of simulations. This is because TXA would be cost-effective if it resulted in any mortality benefit, and the CI for the all-cause mortality treatment effect observed in the HALT-IT trial (95% CI 0.92 to 1.16), does not rule out such a finding. These results also highlight that if a modest treatment effect for a subset of patients can be identified in the future, then TXA has considerable scope to be cost-effective for this group.

## Author affiliations
[1]London School of Hygiene & Tropical Medicine, London, UK
[2]London School of Economics and Political Science, London, UK
[3]Department of Health Services Research and Policy, London School of Hygiene & Tropical Medicine, London, UK
[4]Clinical Trials Unit, London School of Hygiene & Tropical Medicine, London, UK
[5]Department of Medicine, Division of Gastroenterology, Western University, London, Ontario, Canada

**Contributors** All authors (NB, AM, IR, HS-S, VJ and JW) contributed to the study conception and design. The economic model was developed by NB and JW. The economic analyses were performed by NB, AM and JW. All authors (NB, AM, IR, HS-S, VJ and JW) were involved in the interpretation of the study results. The first draft of the manuscript was written by NB. All authors (NB, AM, IR, HS-S, VJ and JW) provided critical review of the manuscript, and have read and approved the final manuscript. JW is the guarantor for the overall content of the article.

**Funding** This work was supported by the UK National Institute for Health Research Health Technology Assessment Programme (HTA/11/01/04).

**Competing interests** None declared.

**Patient and public involvement** Patients and/or the public were involved in the design, or conduct, or reporting, or dissemination plans of this research. Refer to the Methods section for further details.

**Patient consent for publication** Not applicable.

**Ethics approval** The HALT-IT trial was approved by the UK NRES Committee East of England (reference number 12/EE/0038), and by the national and local research ethics committees in all participating non-UK countries. Participants gave informed consent to participate in the study before taking part.

**Provenance and peer review** Not commissioned; externally peer reviewed.

**Data availability statement** All data relevant to the study are included in the article. For the HALT-IT trial data, individual de-identified patient data, including data dictionary, will be made available via our data sharing portal, The Free Bank of Injury and Emergency Research Data (freeBIRD) website indefinitely (https://freebird.lshtm.ac.uk/). The economic model is available from the authors upon request.

**ORCID iDs**
Vipul Jairath http://orcid.org/0000-0001-8603-898X
Jack Williams http://orcid.org/0000-0002-1331-387X

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
