## [Reviewer comments · BMJ Open]

ARTICLE DETAILS

TITLE (PROVISIONAL)	An economic evaluation of tranexamic acid for the treatment of acute gastrointestinal bleeding: A cost-effectiveness analysis using data from the HALT-IT randomised controlled trial
AUTHORS	Bazeer, Nuha; Miners, Alec; Roberts, Ian; Shakur-Still, Haleema; Jairath, Vipul; Williams, Jack

VERSION 1 – REVIEW

REVIEWER	Durand-Zaleski, Isabelle University of Paris
REVIEW RETURNED	22-Jan-2022

GENERAL COMMENTS	Thank you for allowing me to review this economic evaluation of TXA in gastro intestinal bleeding. 1. This economic evaluations is trial-based with a model based extrapolation. I am not sure how the publication of the HTA report, which has basically the same text, tables and results can allow this publication, but I defer to the editors.2. The Markov model chosen to extrapolate the HALT IT results is somewhat over simplistic since it only has 'alive' and 'dead' states, whereas the HALT IT publication indicated a number of adverse events in both groups (Acute MI, stroke, renal failure, liver failure) and many of these events can be expected to have long term effects. The fact that no statistical difference was observed between groups for these events does not preclude including them in the model, since the same absence of difference was also observed for mortality. This is my major comment, I quite disagree with the model as it is, since it does not accurately reflect the condition of patients at the end of the 28-day trial.3. The utility value attributed to the alive state is based upon the TRIGGER trial data. I noted however that the 28-day mortality in TRIGGER was twice lower than the 28-day mortality in HALT IT, which suggest that the patients could be different. Also the TRIGGER trial had fewer complications among survivors, which makes it difficult to ascertain whether the use of TRIGGER data is appropriate.4. Similarly for costs, the figures chosen to extrapolate trial results do not differentiate patients with and without complication at discharge.5. It would be nice to have the result for the trial-based part of the cost calculations, 1) to better understand which part of the total cost is based upon actual prospective data and which part extrapolated,
---

	and 2) also to compare with the TRIGGER cost data. Total 28-day costs in TRIGGER were £2851 (£225), I would expect costs here to be higher given the higher number of procedures and complications. 6. The wording of the discussion and conclusion is a bit circumvoluted because the results are the opposite of what usually happens for drug interventions. The use of TXA appears to be decrementally cost effective but it is unlikely that physicians would add it just to. Potentially reduce costs. I find it strange to calculate and ICER for the reference strategy (no TXA) Minor points  1. The introduction duplicates part of the Lancet 2020 publication and may be shortened 2. The HALT IT trial randomized 12,009 patients but the primary outcome data analysis, which the model used for inputs, was based on data from 11,937 patients, this is probably the figure that should be cited 3. I think the authors' statement that they performed 'a thorough economic analysis using high quality data' is too empathic.
--	---

REVIEWER	Gourzoulidis, George Natl Sch Publ Hlth, Department of Health Services Organization & Management,
REVIEW RETURNED	25-Jan-2022

GENERAL COMMENTS	 1) Introduction: As the main objective of this study is to analyse cost-effectiveness and effectiveness, please elaborate on the same in the introduction section. 2) Mention the economic evaluation guideline utilized 3) Suggest to update the cost in 2022 year 4) I found the discussion part is quite poorly written about the limitations of the model/study 5) table with patient characteristics 6) In the text, the authors report some clinical data about the performance of these two strategies for this patient population. Please, could the authors stress the clinical utility of each strategy by elucidating other clinical index that could improve this section? In my opinion, clinical benefit may represent the first goal of the study and should be clarified by the authors 6) suggest to reported and the breakdown cost in table 3 not only the total cost and QALY
---

VERSION 1 – AUTHOR RESPONSE

REVIEWER 1 COMMENT 1

This economic evaluations is trial-based with a model based extrapolation. I am not sure how the publication of the HTA report, which has basically the same text, tables and results can allow this publication, but I defer to the editors.

We were explicit with the journal that this is the case, and are happy to leave this to an editorial decision.

COMMENT 2

The Markov model chosen to extrapolate the HALT IT results is somewhat over simplistic since it only

has 'alive' and 'dead' states, whereas the HALT IT publication indicated a number of adverse events in both groups (Acute MI, stroke, renal failure, liver failure) and many of these events can be expected to have long term effects. The fact that no statistical difference was observed between groups for these events does not preclude including them in the model, since the same absence of difference was also observed for mortality. This is my major comment, I quite disagree with the model as it is, since it does not accurately reflect the condition of patients at the end of the 28-day trial.

Thank you for this comment. We appreciate the importance of including the adverse events, and accept that some of the long-term outcomes may differ amongst those with and without adverse events.

Firstly, we feel that we should clarify that the costs associated with adverse events which occur within the first 28 days (i.e. the trial period) are included in the model. These costs are included in the hospital length of stay, the hospital procedures that occur, and the transfusions that people receive. However, we realise that this was not clear, so we have amended the text to say:

“Whilst the model did not formally account for adverse events during the trial period, the costs of these adverse events are included in the hospital length of stay, procedures and transfusions that occur during the hospital stay.”

Furthermore, the long-term risk of death data used in the model already includes patients with complications and adverse events. The SMR used in the first year post GI bleed (5.2) and >12 months after GI bleed (1.7) reflect the increase in risk of respiratory, circulatory, and neoplasm risk of death, in addition to GI neoplasm and digestive risks of death. The mortality rate amongst those in the this study (~12%) was similar to that of the HALT-IT trial, so we believe that it is likely to be reflective of the HALT-IT population. As well as the adverse events being similar across both arms, other outcomes are very similar, such as the the Katz score for self-capacity at the end of the trial period. Based on this, we believe that it is appropriate for these long-term risks of death to be the same for both treatment arms at the end of the trial period.

To address the main concern of the comment however, we do appreciate that there is considerable uncertainty amongst the long-term outcomes beyond the trial period, for this patient population. We have added a sensitivity analysis considering a lower utility post-discharge (which addresses the comment below). Whilst the results do impact the ICER (now shown in Table 4), the cost-effectiveness decision does not change across any of these scenarios, and it remains the case that tranexamic is not cost-effective in this indication.

We have also removed two sensitivity analyses that assume that individuals may recover to a general population mortality after 1 year post-GI bleed, and that the utility could return to general population levels, as we appreciate in your comment that there are significantly more comorbidities in this population and that these scenarios may be misleading since they are unlikely they are to occur.

Finally, we have added a point into the discussion to clarify that the model does not formally include adverse events, but that the data used includes those with similar co-morbidities and adverse events:

“Whilst the analysis calculated the hospital costs associated with the hospital stay and procedures performed, the model did not explicitly include the long-term impacts of adverse events. Instead, the data used to extrapolate the costs and effects following the 28-day trial period were assumed to include AGIB patients with similar co-morbidities and complications following their GI bleed.”

COMMENT 3

The utility value attributed to the alive state is based upon the TRIGGER trial data. I noted however that the 28-day mortality in TRIGGER was twice lower than the 28-day mortality in HALT IT, which suggest that the patients could be different. Also the TRIGGER trial had fewer complications among survivors, which makes it difficult to ascertain whether the use of TRIGGER data is appropriate.

Thank you for this comment. This is a very good point, and we appreciate that the patient population in the TRIGGER trial had less complications compared to those in the HALT-IT trial. The utility values from the TRIGGER study were the most suitable values identified in the literature, and therefore we have decided to keep these in the base case analysis. However, we have clarified in the text that the TRIGGER population are likely to have a lower probability of complications upon discharge, and therefore are likely to have a higher utility compared to the HALT-IT population.

We have also performed an additional sensitivity analysis to address the impact of this lower utility, considering a much lower value of 0.5, instead of the higher value of 0.735 used in the base case analysis. These results have been added to Table 4, with other deterministic sensitivity analyses).

“Since the TRIGGER RCT population has a slightly lower level of co-morbidities and complications amongst participants compared to the HALT-IT population, we considered the impact of a much lower

utility value (0.5) for all surviving patients following AGIB.”

The sensitivity analysis demonstrates that there is some uncertainty around the model outcomes. However, given the nature of the cost-effectiveness results (i.e. tranexamic acid reduces overall survival after 28 days), the utility amongst survivors from the trial does not impact upon the cost-effectiveness conclusion that tranexamic acid is not cost-effective. As such, we feel that whilst the exact costs and effects associated with treatment may be uncertain, the conclusion around cost-effectiveness is not uncertain.

COMMENT 4

Similarly for costs, the figures chosen to extrapolate trial results do not differentiate patients with and without complication at discharge.

As stated in our response to comment 2, the costs of the adverse events are considered but only within the 28-day period.

The data source used to estimate the long-term costs and effects beyond the trial period also include patients who have complications however, so on average, these costs are reflective of the overall cohort (i.e. patients with and without adverse events).

The reason that we have not differentiated the costs for those with and without complications is because this would require different data sources which separated out costs based on complications. Similarly to the above points, we did not feel that it would be appropriate to cost each and every possible adverse outcome and include this as a state within the model. Whilst this is a simplifying assumption, we believe it is appropriate, especially given that this would not impact upon the cost-effectiveness decision based on the existing sensitivity analysis results.

COMMENT 5

It would be nice to have the result for the trial-based part of the cost calculations, 1) to better understand which part of the total cost is based upon actual prospective data and which part extrapolated, and 2) also to compare with the TRIGGER cost data. Total 28-day costs in TRIGGER were £2851 (£225), I would expect costs here to be higher given the higher number of procedures and complications.

Thank you for this comment. We have amended the presentation of the results, to show the costs associated with i) TXA treatment and administration ii) hospital based costs within 28 days and iii) post-discharge (i.e. extrapolated) costs. These results are presented in Table 3.

We can confirm that the 28 day costs in this population were approximately £5,112 and £5,162 for tranexamic acid and placebo, which is considerably higher than the £2851 reported by the TRIGGER dataset. We are therefore confident that the costing analysis performed in the model for the 28 day period is accurate and does reflect a population with more comorbidities.

COMMENT 6

The wording of the discussion and conclusion is a bit circumvoluted because the results are the opposite of what usually happens for drug interventions. The use of TXA appears to be decrementally cost effective but it is unlikely that physicians would add it just to. Potentially reduce costs. I find it strange to calculate and ICER for the reference strategy (no TXA)

Thank you. We have edited various parts of the results, discussion and conclusion to emphasise that TXA is not cost-effective, and to try to explain this more clearly.

In the results we have added more explanation of the ICER:

“Since TXA was less costly and less effective in the base case analysis compared to placebo, the ICERs can be reported as £1,576 saved for each LY lost, and £2,209 saved for each QALY lost. Since we value a QALY at £20,000 using the NICE willingness to pay threshold, this does not represent a cost-effective option, because the value of the health outcomes foregone are higher than the cost savings which occur. Alternatively the results can be expressed as no-TXA versus TXA, with no-TXA costing £1,576 per LY gained and £2,209 per QALY gained, meaning no-TXA is preferred at the NICE willingness to pay threshold. Either way, providing TXA is not cost-effective. The deterministic and mean probabilistic results are very similar.”

In the conclusion we have also added that treatment is not recommended under current evidence:

“...TXA administration for patients with AGIB is, on balance, unlikely to be cost-effective at UK WTP thresholds, and should not be recommended based on the current economic evidence.”

COMMENT 7

Minor points

1. The introduction duplicates part of the Lancet 2020 publication and may be shortened

We feel that it is important to provide some background to the decision problem and the context in which this takes place, given this analysis will be a stand-alone manuscript.

In particular, we felt that it was important to present the results of the HALT-IT trial as this is the basis for the decision problem. We also felt that it was important to discuss the meta-analysis publications prior to the HALT-IT trial because these are important to the sensitivity analysis performed. We have tried to strike a careful balance between introducing the relevant aspects of the decision to be made, whilst avoiding repetition.

COMMENT 8

2. The HALT IT trial randomized 12,009 patients but the primary outcome data analysis, which the model used for inputs, was based on data from 11,937 patients, this is probably the figure that should be cited

Thank you. We have now edited the text in the abstract, highlights, and introduction to state that 11,937 patients were included in the analysis.

COMMENT 9

3. I think the authors' statement that they performed 'a thorough economic analysis using high quality data' is too empathic.

We have removed 'thorough' based on your comment, to state: "an economic analysis using high quality RCT data".

REVIEWER 2

COMMENT 1

Introduction: As the main objective of this study is to analyse cost-effectiveness and effectiveness, please elaborate on the same in the introduction section.

The effectiveness of tranexamic acid in this population has been reported in a separate paper in the Lancet ([doi.org/10.1016/S0140-6736\(20\)30848-5](https://doi.org/10.1016/S0140-6736(20)30848-5)). Whilst the effectiveness of the treatment was an important aspect of the economic model, hence why it has been emphasized in the introduction, the aim of this research was to address the decision around the cost-effectiveness of treatment.

COMMENT 2

Mention the economic evaluation guideline utilized

We have now explicitly stated that the economic evaluation followed NICE guidelines. This appears in the 'analysis' section of the methods:

"In line with NICE guidelines for health technology appraisals..."

The NICE guidelines are also mentioned when discussing the willingness to pay threshold and discount rates.

COMMENT 3

Suggest to update the cost in 2022 year

We have now updated the analysis with costs inflated to 2021, the last year within which NHS cost inflation indices are available. We have clarified this in the abstract and the methods, and provided a reference for the inflation index.

"All costs are reported in 2021 Great British Pounds, with costs inflated using an NHS cost inflation index"

COMMENT 4

I found the discussion part is quite poorly written about the limitations of the model/study

Thank you for this comment. We have amended this part of the discussion and discuss some of the limitations of the model structure in more detail.

COMMENT 5

table with patient characteristics

A full table including the main patient characteristics is in the Lancet paper which covers the patient population and clinical effectiveness of tranexamic acid in this population. We have avoided repetition with the Lancet paper (as per comments from reviewer 1), and have decided to only include the characteristics where they are included in the economic model analysis. If the editors feel that it is reasonable to provide a table of the patient characteristics in the appendix, then we will be happy to do so.

COMMENT 6

In the text, the authors report some clinical data about the performance of these two strategies for this patient population. Please, could the authors stress the clinical utility of each strategy by elucidating other clinical index that could improve this section? In my opinion, clinical benefit may represent the first goal of the study and should be clarified by the authors

The clinical effect of tranexamic acid has been explored in more detail in a separate publication of the trial, in the Lancet publication. We are unsure exactly which clinical index would be considered with regards to the cost-effectiveness analysis.

Comment 7

suggest to reported and the breakdown cost in table 3 not only the total cost and QALY

We have now provided a breakdown of the costs as we believe this will be useful for readers to ascertain at which point these costs were incurred. These are presented in Table 3

VERSION 2 – REVIEW

REVIEWER	Durand-Zaleski, Isabelle University of Paris
REVIEW RETURNED	26-Mar-2022

GENERAL COMMENTS	Thank you for your detail answers to each comment and for the changes made in the article. The discussion section addresses adequately the limitations of the model. I have no further comment. I suggest to use only the expression 'cost effective' (or 'not effective') instead of 'the most cost effective' since only 2 options are compared.
--

REVIEWER	Gourzoulidis, George Natl Sch Publ Hlth, Department of Health Services Organization & Management
REVIEW RETURNED	28-Mar-2022

GENERAL COMMENTS	Suggest to use the updated veriosn of CHEERS https://www.bmj.com/content/376/bmj-2021-067975
---

VERSION 2 – AUTHOR RESPONSE

We would like to thank the reviewers for their kind responses and additional comments. We have now addressed both of these comments.

Firstly, we have removed text referring to 'the most cost-effective option' and simplified this by describing it as 'cost-effective'.

Second, we have now used the 2022 version of the CHEERS checklist, and have attached this to our submission. The only change to the manuscript that this resulted in was adding a reference for the HALT-IT trial protocol, which has been added into the Introduction.